# Joint Crisis Plan in Mental Health Settings: A Reflective Process More than an Intervention Tool?

**DOI:** 10.3390/healthcare12242532

**Published:** 2024-12-16

**Authors:** Marie-Hélène Goulet, Sophie Sergerie-Richard, Mathieu Dostie, Jean-Simon Drouin, Luc Vigneault, Christine Genest

**Affiliations:** 1Faculty of Nursing, Université de Montréal, 2375 Chemin de la Côte-Sainte-Catherine, Montréal, QC H3T 1A8, Canada; 2Centre de Recherche de l’Institut Universitaire en Santé Mentale de Montréal, 7401 Rue Hochelaga, Montréal, QC H1N 3M5, Canada; 3Faculty of Pharmacy, Université Laval, Pavillon Ferdinand Vandry, 1050, Av. de la Médecine, Quebec City, QC G1V 0A6, Canada

**Keywords:** joint crisis plan, psychiatry, mental health, mixed methods, partnership, collaborative research, psychiatric advance directives

## Abstract

Background/Objectives: Coercion in mental health is challenged, prompting reduction interventions. Among those, the Joint Crisis Plan (JCP), which aims to document individuals’ treatment preferences in case of future de-compensation, regardless of the potential loss of discernment, has been identified as a key path to study. Identified challenges related to its implementation highlight the need to adapt this intervention to the local context. Considering that in Quebec (Canada), the JCP is not widely used, but the scientific literature supports its adoption and corroborates its potential impact on reducing coercion, this study evaluates the feasibility, acceptability, and preliminary outcomes of the JCP among psychiatric and forensic inpatient settings. Methods: A pilot mixed-methods study was conducted through 16 individual interviews with inpatients and healthcare providers, combined with a pre–post analysis of seclusion and restraint use and the violence prevention climate (VPC) based on healthcare providers’ (n = 57) and inpatients’ perspectives (n = 53). Results: Although the challenging implementation of the JCP complicated the evaluation of its effects on seclusion and restraint use, a moderate change (*d* = 0.40) in the VPC was identified based on healthcare providers’ perspectives. Qualitative findings are also insightful to understand the acceptability and feasibility of the JCP use. A tension emerged between the perspectives of inpatients and healthcare providers: while inpatients valued the reflective process of completing the plan, providers focused more on its technical aspects. Conclusions: The results support the integration of the JCP into patient care pathways, as it provides a tool to amplify patients’ voices, promote patient empowerment, facilitate open dialog on alternatives to coercion, and foster more collaborative and humane mental health care.

## 1. Introduction

For individuals with severe mental illness (SMI), psychiatric and forensic settings often turn to coercion as a last resort (involuntary hospitalization and treatment, seclusion, and restraint) to manage the risk of aggression toward others and self-harm. However, worldwide, the legitimacy of these restrictive practices is being challenged, even as their use remains frequent and does not exclusively reflect a last resort [1]. Indeed, a meta-analysis identified that an average of 14.4% of inpatients in mental health settings experience physical and/or mechanical restraint, 15.8% experience seclusion, and 25.6% experience chemical restraint [2]. It is well documented that these measures harm inpatients, healthcare providers, and the organization [3]. The European Committee [4] has also taken a stance on this matter, defining restrictive practices as security measures with no therapeutic justification. The lack of therapeutic benefits has also been corroborated by a Cochrane systematic review [5]. These measures therefore pose an ethical problem and are the subject of international and provincial policies aimed at reducing or even eliminating their use.

Several interventions have been developed and assessed to contribute to efforts to reduce the use of restrictive practices. Among those interventions, many researchers have corroborated the potential of psychiatric advance directives such as the joint crisis plan (JCP) to reduce coercion in mental health services [6,7]. In order to prevent the use of restrictive measures while promoting the involvement of individuals with SMI, the JCP is the most studied and promising form of advance directive in psychiatry [8]. The JCP allows the individual to, jointly with their healthcare team and relatives, formulate requests to prevent and manage a crisis [9]. Based on the concept of shared decision-making [8], it defines the modalities of intervention in case of a crisis by outlining the individual’s treatment preferences (e.g., preferred alternative intervention to coercion), refusal of treatment, as well as crisis triggers and early warning signs [9,10,11,12]. A study by Loubière, Loundou, Auquier and Tinland [13] suggests that the use of JCPs could result in cost savings associated with a decrease in total psychiatric inpatients days/nights. The JCP also has the potential to reduce compulsory admissions [14,15,16]. Moreover, the collaboration and therapeutic relationship between individuals with SMI and healthcare providers are facilitated by the JCP [17] resulting in the reduction in perceived coercion among inpatients [18,19,20]. Since the results of a systematic review highlight that individuals who have completed a JCP are less likely to engage in violent behavior [20], its use has the potential to reduce coercion aiming to control this behavior.

To better understand the influence of the JCP on the reduction in restrictive practices use, the theoretical framework for the study is the Safewards model [21], which aims to reduce the risk of conflict (crises) and containment (coercion) in psychiatric settings. This model identifies six potential sources of crises: staff team, physical environment, outside hospital, patient community, patient characteristics, and regulatory framework. These sources can trigger factors that may lead to a crisis requiring a restrictive practice. The model helps to better understand how the JCP interacts with these trigger sources to identify more specific strategies for preventing the use of restrictive practices. This model has also influenced the development of scale measuring concepts such as the violence prevention climate [22].

A pre-implementation study conducted in a Quebec (Canada) hospital found that 84.8% of individuals with SMI and healthcare providers surveyed believe that the JCP would meet a critical need in the province’s care settings if it were implemented [23]. According to the participants, the JCP would align with the values, beliefs, and needs of individuals with SMI and healthcare providers. However, barriers such as a lack of organizational support, insufficient training, and the perceived complexity of the intervention could hinder the implementation of the JCP and its anticipated benefits [24]. Therefore, it is recommended that the JCP be adapted to the context in which it will be implemented before considering its broader use in Quebec (Canada) where it is not widely used and where healthcare providers are less familiar with it [25]. Given the challenges related to the implementation of the JCP, but also the scientific literature supporting its adoption and corroborating its potential impact on reducing the use of restrictive practices, it is essential to conduct a pilot study to develop and implement an evidence-based JCP.

This study aims to evaluate the feasibility, acceptability, and preliminary outcomes of the Joint Crisis Plan among individuals with SMI in psychiatric settings. The specific objectives are as follows:To explore the feasibility, acceptability, and preliminary outcomes of the intervention from the perspectives of inpatients and healthcare providers.To compare the prevalence of seclusion and restraint use, and the average score of the violence prevention climate among inpatients before and after the implementation of the JCP.

The hypotheses are that there will be a reduction in the use of coercion (seclusion, restraint), and an increase in the average score of the violence prevention climate.

## 2. Materials and Methods

### 2.1. Study Design

To achieve these objectives, a pilot study with a simultaneous mixed-methods design and triangulation [26] was conducted, where qualitative (descriptive study) and quantitative (single-group pre–post test study) experiments were carried out in parallel. During data analysis, the findings were compared, triangulated, and validated, leading to a richer interpretation of the data [27] and, consequently, a better understanding of the acceptability, feasibility, and preliminary outcomes of the JCP. More specifically, this study has followed the recommendations of Sidani [28] on the development and evaluation of nursing interventions, covering the phases of development, implementation, and evaluation of the intervention.

### 2.2. Study Setting and Participants

The study setting includes the psychiatric emergency department, the psychiatric intensive care unit, a psychiatric acute care unit, and a forensic unit. These are all closed wards that serve patients from an urban city’s area of the province of Quebec (Canada). This mental health institute provides services in several highly specialized areas and serves a population of over 500,000 residents.

### 2.3. Development

To address the challenges associated with translating research findings into practice, this study is based on the principles of organizational participatory research [29]. By prioritizing collaboration among service users, relatives, clinicians, decision-makers, and researchers, this approach allows all key stakeholders to participate meaningfully in the study. Organizational participatory research thus enables the development of projects that address clinical concerns, which would facilitate their implementation and sustainability, thereby bridging the gap between research and practice [30,31].

Then, to develop the JCP and adapt it to the setting, an advisory committee was formed. This committee included two patient partners, a family member, managers and assistant head nurses from the four study units, as well as a representative from the nursing, mental health, and crisis center departments of the study setting, along with the research team. Meetings with the advisory committee allowed for the identification of the essential components of the JCP through a scoping review [9]. Based on the reviewed JCPs, the one developed and implemented by Ferrari’s team in Switzerland was chosen by the committee as the preferred tool to form the basis of the JCP adapted to the study setting. The adapted JCP is available in the Appendix A. The meetings also facilitated the planning of the implementation and monitoring of the intervention.

### 2.4. Implementation

The training program for healthcare providers included two components: one online and the other in person. The online portion (approximately 90 min) aimed to explain the theoretical foundations of the JCP, such as the concept of crisis, the shared decision-making approach, and the steps to complete it jointly. Interactive video modules were developed with the research team, a patient partner, and a family partner. The in-person training, facilitated by the research team, lasted 90 min, and focused on practicing the JCP using clinical scenarios and discussing anticipated barriers and facilitators to its use.

Following the training, a one-month trial period was conducted, during which the research team was present several times a week to answer the healthcare providers’ questions and support the team in the appropriation process.

### 2.5. Evaluation

#### 2.5.1. Qualitative Evaluation

To address Objective 1, semi-structured individual interviews were yielded to gain an in-depth understanding of the perceived feasibility, acceptability, and outcomes of the JCP. Recruitment aimed at 10 healthcare providers and 10 inpatients. The main inclusion criterion for participation was that both the patients and the providers must have completed at least one JCP. The purposive recruitment aimed to identify participants with diverse sociodemographic and clinical characteristics. Providers did not need to be paired with a specific inpatient, as we were not seeking to compare their perspectives.

The selected patients and providers were contacted by a research coordinator who was different from the person responsible for the training. The semi-structured individual interviews were conducted by the person responsible for the training of the inpatient participants, and by the research coordinator for the provider participants. They took place in a private room convenient for the participant and were recorded digitally. The interview guide was based on themes related to the Safewards model [21] and the climate of violence prevention [22,32] for the preliminary outcomes component. The interview guides were validated by a person with experiential knowledge of restrictive practices to assess their completeness, clarity, and relevance.

#### 2.5.2. Quantitative Evaluation

To address Objective 2, for the indicator related to the prevalence of restrictive practices, the convenience sample consisted of patients hospitalized in the targeted care units one year before and six months to one year after the implementation of the JCP depending on the unit. For the psychiatric acute care unit and the psychiatric emergency department, the pre-test sample consisted of inpatients from 1 June 2021 to 31 May 2022, and the post-test sample consisted of inpatients from 1 August 2022 to 31 July 2023. Regarding the forensic unit and the psychiatric intensive care unit, the pre-intervention period is the same, while the post-test sample period extends from 1 August 2022 to 31 March 2023, as a new documentation system was implemented on this date. The intervention was implemented in all units from 1 June to 31 July 2022. Anonymous information related to restrictive practice indicators (restraint and seclusion) was collected from the institution’s electronic registry.

For the indicator related to the violence prevention climate, the modified violence prevention climate scale is a self-administered scale for patients and healthcare providers that aims to measure the violence prevention climate in mental health and forensic settings [32]. This scale was developed to measure the impact of interventions aimed at preventing violence and restrictive measures in general, and forensic psychiatry. In another study [32], we validated and culturally adapted the French-translated version for Quebec based on the violence prevention climate scale [22] with 308 patients and healthcare providers from four university institutes with a mission in mental health and/or forensic psychiatry, including the site identified for the current study. The scale comprises 23 statements covering three factors: staff actions, patient actions, and the therapeutic environment. A Cronbach’s alpha ranging from 0.69 to 0.89 was obtained for the internal consistency of the scale. All patients in the studied units and the healthcare providers working there were invited to complete the French version of the VPC and a sociodemographic questionnaire before (March to April 2022) and after the implementation of the JCP (July to December 2022).

### 2.6. Data Analysis

#### 2.6.1. Qualitative Data Analysis

For the qualitative data, the interviews were recorded, transcribed verbatim, and integrated into NVivo qualitative data analysis software (QSR International, Burlington, MA, USA). A thematic content analysis was conducted for each participant group (inpatients and healthcare providers) involving three activities: data condensation, data display, and data verification [33]. Data condensation was carried out through mixed coding: emergent codes and codes derived from the Safewards model. For data display, summary sheets were produced for each type of participant to identify commonalities and divergences across the different participant groups and codes. For data verification, an inter-coder agreement (involving the research team and an individual with experiential knowledge (LV)) was performed to ensure the reliability of the analysis process.

For the quantitative data, descriptive statistics were used to examine the characteristics of the collected data. All quantitative analyses were performed using IBM SPSS Statistics 29.0.1.0 software.

#### 2.6.2. Quantitative Data Analysis

*Restrictive practices.* The number of seclusion and restraint episodes, as well as the duration of these episodes, were measured daily in both the pre- and post-intervention phases. Time-series analyses were performed to organize the data on a weekly basis [34]. These were designed to evaluate the intervention’s effectiveness on the prevalence and duration of seclusion and restraint episodes in each care unit, each with its own specific characteristics. An expert modeler was consulted to develop ARIMA (p, d, q) models from the time series. The parameters of these models were then manually adjusted to explore other possible configurations and select the models best suited to the data. The values of the normalized Bayesian information criterion (BIC) and the stationary R-squared were considered to evaluate model fit. Ljung–Box tests were conducted to assess stationarity, with a non-significant result expected to confirm the model’s adequacy in terms of residual independence. Autocorrelation function (ACF) and partial autocorrelation function (PACF) graphs of the residuals were also analyzed to verify the absence of residual correlations. ARIMA models with statistically significant parameter estimates were preferred. The final ARIMA parameters selected, and the autocorrelations graph are available in the Appendix A. Statistical analyses were conducted in collaboration with a statistician from the University of Montreal.

*The Violence Prevention Climate.* The French version of the VPC was administered to two independent samples of hospitalized patients in the study setting, both in the pre-test and post-test phases. Shapiro–Wilk tests revealed non-significant results, suggesting a normal distribution of the obtained scores. Therefore, Student’s *t*-tests were used. This same scale was also administered to two samples of staff members in the study setting, both pre- and post-intervention. Since the samples were dependent (two staff members participated in each study condition), effect size calculation was prioritized and measured using Cohen’s d.

For the integration of qualitative and quantitative data, summary tables juxtaposing the two types of data were used to compare qualitative and quantitative results. These were discussed with the research team, including a patient research partner with lived experience on the study topic (LV), to explore different interpretation avenues, focusing on similarities and differences between the results.

### 2.7. Ethical Considerations

The study was approved by the Ethics Committee of the hospital where the study was conducted (Reg. No. 2021-2458). Before the interviews began, we explained the purpose of the study to each participant both verbally and in writing. They all signed a consent form before the start of the interview. We informed them that we required their consent to participate in the study, that they could withdraw at any time (including after the interview), and requested their permission to use a voice recorder. To acknowledge their essential contribution to this study and the time dedicated to the interviews, a $20 gift card was provided to the inpatient participants as compensation. Healthcare providers participated in the interviews during their work hours. Furthermore, to protect the confidentiality of the interviewees, we assigned a code to each participant and removed any identifying information from the transcripts. The recordings have been transcribed and subsequently destroyed. According to the guidelines of the ethical committee, these research data will be retained for at least 7 years by the principal investigator of this research project (MHG).

## 3. Results

First, from the four initially targeted units, the psychiatric emergency department staff was more reluctant to use the JCP, resulting in no implementation in this unit. The forensic unit adopted it specifically as a planned discharge tool, while the manager of the psychiatric intensive care unit identified the JCP as a tool that needed to be completed for every new admission. Its use in the psychiatric acute care unit varied depending on the staff.

The qualitative results are presented first, followed by the quantitative results. The integration of these two components is addressed in the illustration below (see Figure 1) and in the Discussion Section.

### 3.1. Qualitative Results

We conducted sixteen interviews: eight with healthcare providers (six nurses and two specialized educators) and eight with inpatients. Interviews with healthcare providers lasted approximately 45 min, while those with inpatients lasted around 35 min.

Among the healthcare providers, five identified as female and three as male. Four were aged between 29 and 38 years, three were aged between 54 and 56 years, and one did not disclose their age. Their work experience in mental health varied, with three having less than 10 years, three having 13 to 14 years, and two with over 25 years of experience. Two healthcare providers did not specify their unit, while two were from the psychiatric acute care unit, two from the forensic unit, and two from the psychiatric intensive care unit. Among the inpatients, five identified as male and three as female. One was under 20 years of age, two were between 20 and 25, five were between 30 and 40, and one was over 40 years. Diagnoses reported by inpatients included schizophrenia (n = 2), bipolar disorder (n = 4), and borderline personality disorder (n = 1), with one individual not disclosing their diagnosis. Three inpatients were hospitalized in the psychiatric intensive care unit, three in the forensic unit, one in the psychiatric acute care unit and one did not specify their unit. 

In order to address the first objective, the results are divided into three themes: (1) the hospital context, (2) the tension between the tool and the reflective process, and (3) a helpful tool giving a voice to inpatients. The first two themes focus on the feasibility and acceptability of the JCP and the third one on the perceived outcomes.

#### 3.1.1. Theme 1: The Hospital Context

The three units where the JCP was implemented (forensic, psychiatric intensive care, and psychiatric acute care) are characterized by distinct patient populations, but participants shared recurring elements regarding the influence of the psychiatric hospital environment on JCP use. The hospital context encompasses all the attributes of the hospital environment, including, for example, administration and care routines. Specific characteristics of this context have been identified as influencing feasibility and acceptability based on participants’ perceptions.

In terms of workload, the JCP could be seen by some healthcare providers as an additional task, and as an additional tool to an already extensive array of documents to complete. Some healthcare providers felt that they were discussing the same elements included in the JCP through other tools such as the post-incident debriefing or the safety plan. Indeed, they shared that there are already several tools aimed directly or indirectly at preventing crisis situations. However, for other providers, the JCP was seen as complementary to the other tools, which are mostly completed by the healthcare providers rather than the inpatients themselves. This workload, which translates into a lack of time, leads to a limited use of the JCP, as some providers report that even though some JCPs were completed, they were not necessarily consulted: “*I would say that even if the document is completed, I feel that few nurses have taken the time to read it and ensure that it accurately reflects what the patient wants and how they want it. […] In the heat of the moment, there is this practice, this routine* [to be developed].” (Nurse 2). Although the tool was perceived as time-consuming and could represent an additional task, this same nurse mentioned: “*You might spend 20 min now, but it might be 20 min that will prevent four assaults, so it’s not a waste of time.*” This quote highlights the preventive aspect of the JCP, which requires time but aims to prevent assaults that would otherwise increase the workload.

The length of hospital stays also emerged as a factor to consider in the hospital context. On the units designated for short-term hospitalizations where there are many admissions, such as the psychiatric acute care unit, healthcare providers mentioned that this constrained the completion of the JCP or its use once completed due to the limited time spent with the patient. For inpatients, this is important because the acceptability of the JCP largely depended on its use, once completed, outside the hospital. Indeed, inpatients expressed a desire to have long-term access to it once they are at home or in supervised housing, to prevent further events that could lead to hospitalization. One participant shared: “*I plan to keep my copy somewhere visible so I can look at it more often*” (Patient 1).

Favoring the use of the JCP outside the hospital was also reflected in certain questions that seemed more relevant once discharge had been granted even if the discharge should be planned as early as the admission. One person provided an example with the section of the JCP concerning the identification of professional resources to contact if needed: “*My professional resources… it’s more them here at the hospital… it could be any orderly or nurse […] I wasn’t sure what to fill in there*” (Patient 2). In the hospital context, the patient does not necessarily choose the available professional resources, whereas outside the hospital, this question allows for the identification of community-based providers, for example. 

Thus, the hospital environment, characterized by a heavy workload and variable lengths of hospitalization, represents a complex context that influences the feasibility and acceptability of the JCP, according to most of the participants.

#### 3.1.2. Theme 2: The Tension Between the Tool and the Reflective Process

This theme addresses the distinction between the perspective of healthcare providers, who primarily view the JCP as an intervention tool once completed that they can rely on to prevent a crisis, and the perspective of inpatients, who focus more on the reflective process that accompanies the completion of the JCP, proposing that completing a JCP is in itself the intervention. This tension is illustrated in Figure 1, where the two perspectives are presented in parallel and in contrast.

Several healthcare providers shared their perspectives, focusing on the use of the JCP and making little reference to the accompanying process, which involves completing and reflecting on the tool together with the patient, which was the emphasis of the training. Many of the healthcare providers’ comments centered on the initiation of the tool’s completion and directly addressed the potential use of the JCP once it was completed.

Regarding the initiation of the completion of the JCP, based on the accounts shared by the providers, inpatients are generally open to completing the JCP, which was surprising for some of them. For the providers, the JCP is a tool that adapts easily, making it suitable for almost all individuals. Contradictory views were also shared on the matter: Some providers cautioned about people with borderline or antisocial personality disorders, citing concerns about the reliability of their responses to the questions. In contrast, one participant working with individuals with antisocial personality disorder found the tool to be quite relevant: “*The approach needs to be different because it’s more difficult for them* [people with antisocial personality disorder] *to admit their difficulties and behavior. So I think you need to build a connection with them and bring it into a more collaborative approach […]. I think if you ask the questions and the answers come from them, I think it’s feasible*” (Specialized Educator 1).

This concern about the reliability of the JCP content is reflected in the providers’ consideration of the appropriate timing for implementing the JCP in inpatient’s care trajectory. These considerations are related to the importance of choosing the right moment based on the individual’s condition. While some providers argued that the timing of the JCP completion should be determined by the patients themselves, others felt that a person in a crisis situation would not be able to start completing the JCP and that it would be better to wait until their clinical condition was more stable. This idea is reflected in the following comment from a nurse working in the psychiatric intensive care unit: “*It’s the place where the person is that is not suitable at 100% for this JCP, so I dare say that a regular department where the person is more stable […] might be able to clearly express how to intervene if they are in crisis*.” (Nurse 3). This concern about the crisis state reflects a worry regarding the application of the JCP. Indeed, some providers feared that asking individuals to identify strategies to prioritize in order to prevent a crisis while they are in a crisis could undermine the reliability of the JCP content and, consequently, its subsequent use by providers.

On the other hand, among inpatients, the discussions place more importance on the reflective process that accompanied the completion of the JCP. They focused on the constructive nature of discussing their past behaviors during crisis situations. Although one person mentioned that the reflective process associated with completing the JCP could lead to feelings of guilt by recalling unpleasant memories, most individuals reported appreciating the reflection time offered by the JCP, as expressed by the following participant: “*It’s just the reflecting part. Like: What makes me go into crisis or like what triggers me is really the reflective question… Even the question about what makes me happy, even the hard ones are helpful*” (Patient 3).

The attention given to the process among inpatients is illustrated by the importance placed on the collaborative nature of the JCP. Based on the accounts of inpatients, the “joint” completion of the JCP varied significantly. While some completed it on their own as requested by the providers without receiving feedback, others completed it with a healthcare provider: “*They still reviewed the sheet, but we hadn’t looked at it together. But that would have been good, to look at it together. […] I still think that by looking at it with them, they might have comments or perhaps feedback on all of it*” (Patient 4). This quote illustrates the perceived added value of undertaking this process jointly with a provider. Only one patient reported completing the JCP jointly with a provider and a close relative.

Thus, from the perspective of inpatients, the JCP primarily serves as a reflective tool that allows for constructive joint completion with providers. For providers, however, the focus was more on starting the JCP at the right time and place to ensure its content is applied, highlighting the tool’s relevance, especially once completed.

#### 3.1.3. Theme 3: A Helpful Tool Giving a Voice to Inpatients

Participants shared several outcomes resulting from the use of the JCP, all agreeing on its helpfulness, as illustrated in Figure 1. These will be presented below, dividing the reported outcomes according to the perspectives of the providers and the inpatients.

Healthcare providers’ perspectives: the only tool that gives a voice to patients

According to providers, inpatients involved in a JCP feel valued, heard, and involved, and they have the sense of having the power to express how they want to be approached by a provider. This leads them to perceive the provider less as a threat.

Some providers specify that the JCP is the only tool, among those used in the hospital setting, that belongs to the patient. However, the tool still allows providers to better understand what the person wants or does not want by offering intervention guidelines. Moreover, from a perspective where the tool follows the individual’s care trajectory, the JCP could even enable tracking the person’s progress and better understanding their journey for the providers.

In the same vein, providers recognize that the JCP has allowed them to open discussions on topics perceived as taboo, such as violence and coercion. One provider shared this reflection: “*I think we all fear being assaulted. No one wants to be assaulted. So maybe discussing topics like talking to patients about their periods of aggression, I think it wasn’t easy to talk about before. Now we can talk about it. […] We can talk directly about the real things that scare us and make us uncomfortable. […] Everything is written out in words* [in the JCP]. That’s good. *That’s what I find good*” (Nurse 4).

According to the providers, two other outcomes have positive effects for both the inpatients and themselves: the provision of a secure, protocol-driven, and structured framework that the JCP allows, and the enhancement of the trust relationship with the patient, as the JCP positively influences the therapeutic relationship.

Inpatients’ perspectives: a genuinely helpful tool

Inpatients also report that the JCP provides an opportunity to put into writing elements discussed informally with the staff, thereby making these exchanges more official. One aspect shared by many is the perception that the JCP is a tool that can genuinely help them, particularly once they are alone outside the hospital. The entire process of completing the JCP is viewed positively because of the constructive reflection time offered by the questions in the form and the discussions with the providers, where everyone’s viewpoints are welcomed.

Among the interview participants, one person was rehospitalized after having completed a JCP during a previous hospitalization. Upon returning to the hospital due to a high risk of suicide, this person mentioned that the JCP helped them seek help more quickly when they needed it: “*I am quicker to reach for help. Compared to the first hospitalization, where it was completely impulsive and didn’t ask anybody for help. The second time, I called the Crisis Center*” (Patient 3). 

According to the inpatients, the JCP allows providers to get to know them better, which reduces their perception of danger that can have an impact on the perceived coercion. One person mentions the following: “*I feel like they are more hesitant to restrain me right away*” (Patient 3).

The JCP also has a positive impact on the relationship with providers, according to the inpatients, in addition to promoting the use of identified personal strategies to avoid coercion.

### 3.2. Quantitative Results

The results related to seclusion and restraint use as well as the violence prevention climate are presented in this section.

#### 3.2.1. Duration and Prevalence of Seclusion and Restraint Episodes

In the psychiatric acute care unit, the weekly duration of seclusion episodes shows a decrease over time according to the time-series data (Figure 2), particularly following the intervention. An ARIMA model (2, 0, 2) was identified, and the intervention parameter (−9097.98) was found to be statistically significant, *t* = −2.14, *p* = 0.035. This result indicates a change after the implementation of the JCP on the time series, specifically a reduction in the duration of seclusion episodes. ARIMA models applied to other time series did not reveal an effect from the intervention on the reduction in prevalence or on the duration of restraint or seclusion episodes. Indeed, a non-significant decrease in the prevalence of seclusion episodes was observed on the psychiatric acute care unit (Figure 3). However, despite being statistically non-significant, a decrease in the prevalence and weekly duration of restrictive measures was observed in the psychiatric intensive care unit (Figure 4 and Figure 5). The descriptive data of each ward are listed in Table 1 and Table 2.

#### 3.2.2. Violence Prevention Climate Among Inpatients

The characteristics of inpatient samples are listed in Table 3. Student’s *t*-tests (one-tailed) for independent samples show no significant results (Table 4). This suggests that the inpatient samples following the implementation of the JCP did not perceive any deterioration or improvement in the violence prevention climate.

#### 3.2.3. Violence Prevention Climate Among Healthcare Providers

The pre-test sample consists of 23 healthcare providers, including 13 nurses (56.52%), while the post-test sample includes 34 healthcare providers, with 15 nurses (44.12%). The results indicate a reduction in the average scores obtained following the implementation of the intervention, suggesting an improvement in the violence prevention climate as perceived by the healthcare providers. Several effect sizes were measured based on the average scores obtained by the healthcare providers samples on the VPC-M-FR scale (*d* = 0.40), the staff actions subscale (*d* = 0.46), the patient actions subscale (*d* = 0.07), and the therapeutic environment subscale (*d* = 0.22) (Table 5). These results indicate moderate differences between the average scores obtained, except for the patient actions subscale, where the difference was found to be very small [35].

## 4. Discussion

This study, aimed at evaluating the feasibility, acceptability, and impact of the JCP on the use of seclusion and restraint, based on the perceptions of inpatients and healthcare providers, highlighted the complexity associated with implementing such an intervention in a hospital setting and the tension between the perceptions of healthcare providers and patients regarding its usefulness. 

Indeed, hospital constraints such as workload and short hospitalization durations present barriers to the use of the JCP. Since we did not interview healthcare providers from the psychiatric emergency unit, we lack insight into the specific reasons they perceived as influencing incomplete implementation in this setting. However, during the study, workload and competing project priorities in this unit were identified by the research team as potential obstacles to its participation. Most studies on JCP implementation have targeted outpatients. In fact, Tinland, Loubière, Mougeot, Jouet, Pontier, Baumstarck, Loundou, Franck, Lançon, Auquier, and Group [16] conducted a randomized controlled trial in non-hospital settings where the JCP was completed jointly with a peer support worker rather than with clinicians. Completion with a peer support worker was associated with greater reductions in involuntary hospitalizations and psychiatric symptoms, as well as increased perceived empowerment and recovery. In light of these results, and considering that the hospital context hinders the use of the JCP, it would be worthwhile to plan future research focusing on JCP implementation in community settings in Quebec (Canada). The flexibility of the JCP, due to its broad questions that can apply in different contexts, allows it to prioritize the individual’s life trajectory rather than their care pathway within healthcare services, thus promoting its use in daily life outside the hospital. Furthermore, Tinland, Loubière, Mougeot, Jouet, Pontier, Baumstarck, Loundou, Franck, Lançon, Auquier, and Group [16] suggest such an implementation could involve peer support workers and encourage family involvement, as only one JCP in our study was completed in the presence of a family member. To facilitate implementation in a new study setting and identify effective implementation strategies from the outset, tools from implementation science could be mobilized to account for the specific characteristics of community-based contexts.

Among the inpatients, only one person reported that the JCP was completed in the context of a suicidal crisis, while participants more often referred to crises involving self-harm or aggressive behavior. Stanley and Brown’s safety plan [36], on the other hand, was specifically developed and studied in the context of suicidal crises. However, the elements included in the safety plan are similar to those addressed in the JCP (i.e., warning signs, coping strategies, strategies for safety). Moreover, some studies have shown that the safety plan has had positive outcomes in reducing hospitalizations and improving psychiatric symptoms in populations not necessarily presenting suicidal behavior. The JCP, or any other crisis plan like the safety plan, proves relevant regardless of the type of crisis, raising questions about the necessity of focusing on the type of crisis when selecting the appropriate crisis plan.

Hudson, Pariseau-Legault, Cassivi, Chouinard and Goulet [37] propose a more holistic definition of a mental health crisis, unique to each individual, where it is the impact on their social, occupational, and interpersonal functioning that determines the blurred boundaries of the crisis. By adopting this view and considering that a crisis is defined by the person’s situation, the JCP can be used in a variety of scenarios, whether the crisis is associated with psychotic, suicidal, self-harming, or aggressive behaviors. This raises an interesting question about the added value of having different plans for different types of crises versus a more generic plan, such as the JCP, which focuses not on the type of crisis but on its characteristics and the strategies for addressing it. Furthermore, considering that patients reported positive outcomes simply from completing the tool, it is again questionable whether to focus on the type of crisis when using a crisis plan like the JCP. The discussion involved in using the JCP ensures a moment of reflection that is as constructive as the completed tool itself. To this end, a positive therapeutic relationship can facilitate the process but can also be an outcome of the joint completion process itself. Given that some inpatients in our study could not benefit from this process because they completed it alone, it highlights the need to emphasize the importance of joint completion in the training on JCP.

In comparison to other studies that highlighted the impact of the JCP on the use of formal coercion (seclusion and restraint) [9], our quantitative results do not point to a statistically significant decrease. Indeed, participants did not spontaneously discuss the reduction in coercion as a possible outcome of the JCP, but certain elements suggest that the JCP may influence perceived coercion. Indeed, healthcare providers were seen less as a threat, there was an openness to discussing sensitive topics such as violence and coercion, and the exchange led to a reduction in the perception of danger. To our knowledge, this is the first study that evaluates the impact of the JCP or other psychiatric advance directives on the VPC. Although the quantitative results do not show a statistically significant reduction in the climate of violence prevention according to patients’ views, healthcare providers were able to better understand the patients following the joint completion of the JCP, which tempered their perception of danger and could have a potential impact on the use of coercive practices. Based on the providers’ perceptions, the results suggest an improvement in the violence prevention climate following the implementation of the JCP. This significant change could be explained by the participatory design of the study, the training they received, or the clinical support provided during the study, all of which emphasized the importance of preventing crises through interventions like the JCP. These efforts may have mobilized factors associated with violence prevention and, ultimately, influenced the overall climate. On the other hand, since inpatients were not aware of these study activities, this might explain why they did not report any change in the violence prevention climate after the implementation of the JCP. 

These limitations in obtaining significant quantitative results underscore the need to question the relevance of the privileged indicators currently used to study the effects of the JCP. Often understood and utilized as a crisis prevention tool, the focus tends to be on clinical indicators such as seclusion, restraints, and involuntary hospitalizations [9]. However, recognizing the potential benefits of the JCP on the recovery process—particularly its reflective and constructive impacts—it may be more relevant to consider other indicators. Rather than solely viewing the JCP as an intervention tool, it could be reinforced as a psychiatric advance directive, empowering inpatients by amplifying their voice and rights within mental health settings. While the long-term effects might include reductions in crises and coercion, the most immediate impacts may be observed in indicators such as perceived recovery, empowerment, self-determination, and therapeutic relationships.

## 5. Limitations

This study, however, has limitations. First, for the quantitative part of the study, we were unable to collect the total number of JCPs completed across the three units where it was implemented, which hindered our understanding of the implementation process of the tool. Also, regarding the variable of violence prevention climate, healthcare providers’ awareness of the project may have biased the results, showing a moderate difference between pre-test and post-test measures. The data collected by the organization regarding the use of restrictive measures only allow for counting the number of restraint and seclusion episodes, without providing the number of patients who experienced formal coercion. Therefore, it is not possible to assess the impact on inpatients who were more frequently subjected to seclusion and restraint, for example. Additionally, chemical restraint was not yet recorded as a restrictive measure at this hospital during the study and is thus excluded from the collected data. However, the data do allow for observation of the duration of episodes. Involuntary hospitalization was also not measured in this study, as these data are collected by an independent system that does not allow for pairing. This is a significant limitation of the study, considering that involuntary hospitalizations represent a key variable influenced by the use of the JCP [15]. Furthermore, although the joint nature of the JCP involves the participation of a relative, only one JCP was completed with a family member involved during the study. Nonetheless, this is an interesting result in terms of the feasibility of involving family members in the context of this study.

Other strengths of the study must be highlighted. Although one unit did not implement the JCP, the involvement of key stakeholders in a participatory organizational approach certainly helped in adapting the JCP to overcome barriers to its use in the other settings. The inclusion of a patient partner at every stage of this study is also a strength, as their lived experience informed the development of the JCP and ensured a more in-depth data analysis.

## 6. Conclusions

In conclusion, this study helps bridge a gap in the literature by evaluating the feasibility and acceptability of the JCP, as well as its impact on the use of seclusion and restraint and the violence prevention climate based on the perceptions of inpatients and healthcare providers. While the results confirm the feasibility and acceptability of implementing the JCP, they also underscore the challenges of introducing such a tool in clinical settings, particularly in hospital environments where numerous barriers exist. The lack of a significant impact on the use of seclusion and restraint, combined with a moderate effect on the violence prevention climate, highlights the need to look beyond these clinical outcomes and focus on the JCP’s potential to enhance the recovery process and therapeutic relationships. The preliminary outcomes suggest that the JCP has significant potential to improve care pathways for individuals living with severe mental illness by fostering constructive self-reflection and strengthening their relationships with healthcare providers. As such, the JCP emerges as a promising tool to promote patient empowerment, encourage open dialog about alternatives to coercion, and support more collaborative and humane mental health care.

## Figures and Tables

**Figure 1 healthcare-12-02532-f001:**
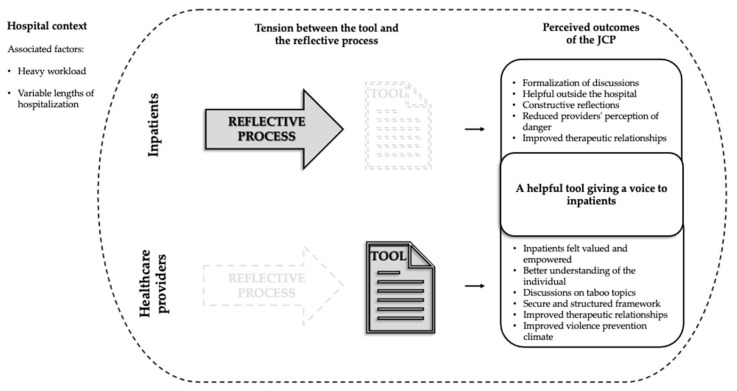
Perceived outcomes of the JCP: navigating the tension between the reflective process and the tool.

**Figure 2 healthcare-12-02532-f002:**
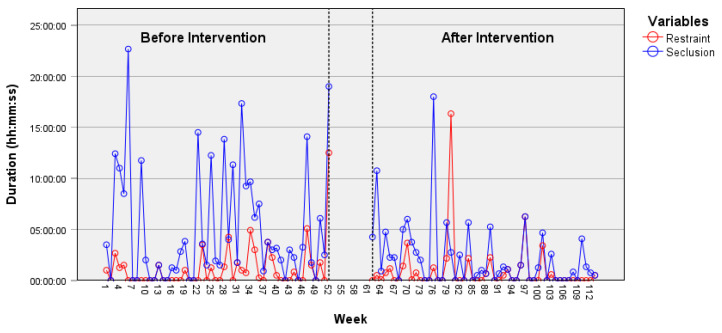
Duration of restraint and seclusion episodes in the psychiatric acute care unit.

**Figure 3 healthcare-12-02532-f003:**
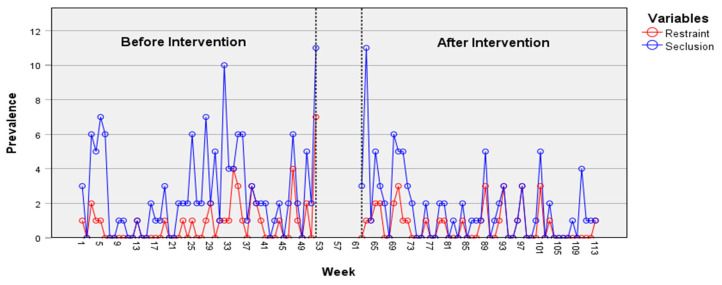
Prevalence of restraint and seclusion episodes in the psychiatric acute care unit.

**Figure 4 healthcare-12-02532-f004:**
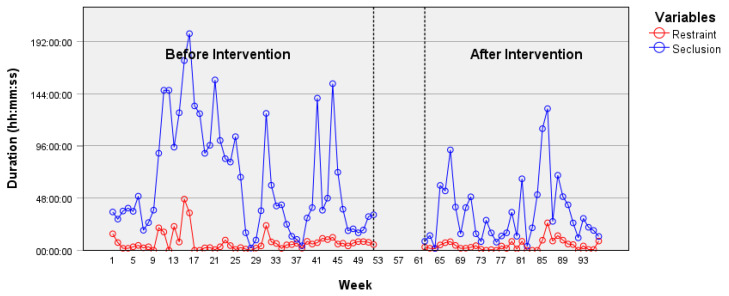
Duration of restraint and seclusion episodes in the psychiatric intensive care unit.

**Figure 5 healthcare-12-02532-f005:**
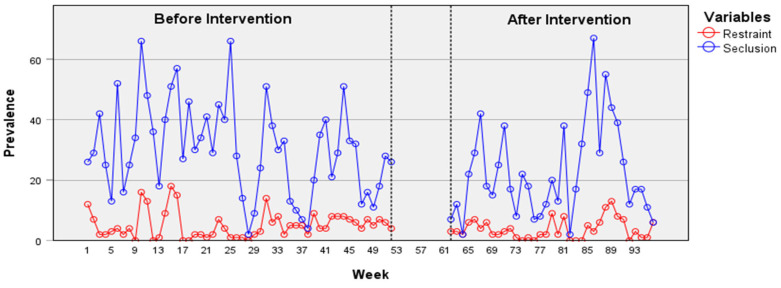
Prevalence of restraint and seclusion episodes in the psychiatric intensive care unit.

**Table 1 healthcare-12-02532-t001:** Descriptive statistics of seclusion episodes.

Ward	Samples	Duration (hh:mm:ss)		Prevalence	
		Median	Mean ± SD	Median	Mean ± SD
Psychiatric acute care	Pre-test	2:55:00	5:03:33 ± 5:43:46	2.00	2.67 ± 2.66
	Post-test	0:57:30	2:10:57 ± 3:13:47	1.00	1.71 ± 2.15
Psychiatric intensive care	Pre-test	40:00:00	65:21:38 ± 51:18:40	29.00	30.21 ± 15.35
	Post-test	25:10:00	34:39:52 ± 30:20:29	18.00	22.74 ± 15.64
Forensic	Pre-test	0:00:00	7:28:45 ± 30:20:08	0.00	1.15 ± 4.09
	Post-test	0:00:00	14:46:17 ± 42:38:40	0.00	1.97 ± 5.47
Psychiatric emergency	Pre-test	5:52:30	8:48:10 ± 8:24:47	4.00	5.33 ± 3.55
	Post-test	5:50:00	8:57:36 ± 9:15:28	5.00	5.00 ± 2.98

**Table 2 healthcare-12-02532-t002:** Descriptive statistics of restraint episodes.

Ward	Samples	Duration (hh:mm:ss)		Prevalence	
		Median	Mean ± SD	Median	Mean ± SD
Psychiatric acute care	Pre-test	0:00:00	1:08:10 ± 2:06:35	0.00	0.85 ± 1.35
	Post-test	0:00:00	0:54:19 ± 2:28:49	0.00	0.65 ± 0.97
Psychiatric intensive care	Pre-test	5:15:00	7:21:50 ± 8:44:48	4.00	5.15 ± 4.37
	Post-test	3:00:00	4:29:26 ± 5:03:14	3.00	3.74 ± 3.35
Forensic	Pre-test	0:00:00	0:21:21 ± 1:29:53	0.00	0.19 ± 0.63
	Post-test	0:00:00	0:10:51 ± 0:53:23	0.00	0.09 ± 0.37
Psychiatric emergency	Pre-test	3:57:30	5:24:14 ± 6:51:16	3.00	3.44 ± 2.64
	Post-test	3:22:30	5:31:21 ± 6:40:46	3.00	3.35 ± 2.27

**Table 3 healthcare-12-02532-t003:** Descriptive statistics of inpatient samples.

Characteristics	Pre-Test		Post-Test	
	N = 18		N = 35	
	Mean ± SD		Mean ± SD	
Age (years)	39.94 ± 13.09		43.14 ± 13.05	
	*n*	%	*n*	%
Gender				
Female	8	44.44	13	37.14
Male	10	55.56	22	62.86
Diagnosis ^a^				
Anxiety disorders	0	0.00	1	2.86
Bipolar disorders	4	22.22	16	45.71
Depressive disorders	2	11.11	1	2.86
Personality disorders	2	11.11	1	2.86
Psychotic disorders	7	38.89	14	40.0
Substance-related disorders	1	5.56	1	2.86
Not specified	2	11.11	1	2.86

^a^ Self-reported diagnosis.

**Table 4 healthcare-12-02532-t004:** Between-group comparisons of VPC-M-FR scale and subscales scores in inpatients pre- and post-test.

Scale and Subscales ^a^	Sample	Median ^b^	Mean ± SD ^b^	t	Mean Difference [95% CI]	*p*
Modified violence	Pre-test	57.5	58.33 ± 8.77	−0.36	−1.27	0.36
prevention climate scale	Post-test	58.0	59.60 ± 13.30		[−8.24; 5.71]	
Staff actions	Pre-test	30.5	32.56 ± 5.92	−0.15	−0.39	0.44
subscale	Post-test	32.0	32.94 ± 10.24		[−5.65; 4.87]	
Patient actions	Pre-test	10.0	10.06 ± 2.71	−0.68	−0.57	0.25
subscale	Post-test	10.0	10.63 ± 2.99		[−2.26; 1.12]	
Therapeutic environment	Pre-test	16.0	15.72 ± 2.78	−0.36	−0.31	0.36
subscale	Post-test	16.0	16.03 ± 2.97		[−2.00; 1.39]	

^a^ Scores for items 8, 12, 15, 17, and 18 have been reverse coded. ^b^ Range: 1–5. Response scale: (1) strongly agree; (2) agree; (3) neither agree nor disagree; (4) disagree; (5) strongly disagree.

**Table 5 healthcare-12-02532-t005:** Descriptive statistics of samples of healthcare providers.

Scale ^a^	Sample	Median ^b^	Mean ± SD ^b^
Modified violence	Pre-test	60.00	57.30 ± 9.11
prevention climate scale	Post-test	53.00	53.59 ± 9.39
Staff actions	Pre-test	25.00	26.74 ± 5.92
subscale	Post-test	24.50	24.12 ± 5.63
Patient actions	Pre-test	13.00	13.17 ± 2.84
subscale	Post-test	12.50	13.00 ± 2.58
Therapeutic environment	Pre-test	17.00	17.39 ± 4.30
subscale	Post-test	16.00	16.47 ± 4.15

^a^ Scores for items 8, 12, 15, 17, and 18 have been reverse coded. ^b^ Range: 1–5. Response scale: (1) strongly agree; (2) agree; (3) neither agree nor disagree; (4) disagree; (5) strongly disagree.

## Data Availability

The data presented in this study are available on request from the corresponding author due to the ownership of the datasets by the institution where the study was conducted.

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
