# Peer review of "Joint Crisis Plan in Mental Health Settings: A Reflective Process More than an Intervention Tool?"

_healthcare, 2024, doi:10.3390/healthcare12242532_

Round 1

Reviewer 1 Report

Comments and Suggestions for Authors

Thank you to review your paper entitled: 'Joint crisis plan in mental health settings: a reflective process 2 more than an intervention tool?'.

This is a very well-written paper on an important topic within mental health care. The introduction is clear, well-substantiated, and the rationale is evident. The methodology is thoroughly described, making the study easily reproducible. One minor remark for completeness in the ethical section: could you elaborate on the data storage process?

In the results section, it is important to avoid interpretations as much as possible, although this can be challenging in qualitative research. Interpretations are particularly prominent in L294, L313, and L432.

Finally, I must note that the discussion lacks alignment with existing research. While you interpret your own results, they are not compared or contrasted with prior studies.

Reviewer 2 Report

Comments and Suggestions for Authors

Dear, it was a pleasure to have contributed to the review of this manuscript, which I found in its content to be extremely timely, complete, and functional. The references are relevant, the English used for editing is understandable and fluent, and the overall structure of the manuscript is complete. There are, however, a few minor criticisms that deserve to be corrected before approving the manuscript:

1) the abstract lacks statistical analysis references (mean and s.dev), as well as sample references.

2) a summary table of literature results should be included in the introduction highlighting the strengths and weaknesses (shortcomings), which then convinced the authors to carry out this work

3) a table should be included at the end of the manuscript, in the discussions, indicating the key points of the search and thus the results;

4) the reading of the results should be streamlined a bit, because it is heavy to read;

5) the limitations of the study should be hived off from the discussions and put in a separate paragraph.

6) the conclusions should be implemented, better arguing the outcome of the results in light of the research objectives.
